# Hybrid kernel polynomial method

Muhammad Irfan,[1] Sathish R. Kuppuswamy,[1] Dániel Varjas,[2, 1, *] Pablo M. Perez-Piskunow,[1, 3]
Rafal Skolasinski,[2, 1] Michael Wimmer,[2, 1] and Anton R. Akhmerov[1]

[1]*Kavli Institute of Nanoscience, Delft University of Technology,*
*P.O. Box 4056, 2600 GA Delft, The Netherlands*
[2]*QuTech, Delft University of Technology, P.O. Box 4056, 2600 GA Delft, The Netherlands*
[3]*Catalan Institute of Nanoscience and Nanotechnology (ICN2),*
*CSIC and BIST, Campus UAB, Bellaterra, 08193 Barcelona, Spain*

The kernel polynomial method allows to sample overall spectral properties of a quantum system, while sparse diagonalization provides accurate information about a few important states. We present a method combining these two approaches without loss of performance or accuracy. We apply this hybrid kernel polynomial method to improve the computation of thermodynamic quantities and the construction of perturbative effective models, in a regime where neither of the methods is sufficient on its own. To achieve this we develop a perturbative kernel polynomial method to compute arbitrary order series expansions of expectation values. We demonstrate the efficiency of our approach on three examples: the calculation of supercurrent and inductance in a Josephson junction, the interaction of spin qubits defined in a two dimensional electron gas, and the calculation of the effective band structure in a realistic model of a semiconductor nanowire.

## I. INTRODUCTION

The behavior of the Fermi sea is governed by both the few partially occupied states near the Fermi level, and the overall effect of the large number of fully occupied states. Therefore, in order to accurately capture the relevant physics, one needs to combine high resolution information about the former with integrated contribution of the latter. A similar need to combine integrated information with high resolution arises when constructing effective models using Löwdin partitioning or Schrieffer-Wolff transformation [1–4]. In a computational context, simultaneously satisfying these two requirements is only possible with the full knowledge of the spectrum. Therefore analyzing a system with size $N$ Hilbert space requires the full cost of $\mathcal{O}(N^3)$ operations of dense linear algebra, prohibiting the exploitation of the sparsity of the Hamiltonian.

Applying a sparse Hamiltonian to a state is cheap. Iterative diagonalization algorithms efficiently utilize this to obtain a small set of eigenenergies and eigenvectors at a low cost [5]. For example, the algorithms implemented in ARPACK [6] combined with a sparse direct linear solver (such as MUMPS [7, 8]) allow to compute several eigenvectors around any interior point of the spectrum. The kernel polynomial method (KPM) [9] also utilizes the sparsity structure, but to obtain limited energy resolution information about the full spectrum. This is possible due to recursive computation of the Chebyshev decomposition of the Hamiltonian action on a vector.

In this work we propose a family of algorithms which we call "hybrid KPM" that combine the integral information of KPM with the high precision of diagonalization. The building block of these methods is the amended KPM expansion, where we subtract the contribution of the known part of the spectrum. Hybrid KPM algorithms apply both to the computation of thermodynamic properties at low temperatures, and the construction of effective Hamiltonians restricted to a small subspace. We demonstrate on a set of physical problems that hybrid KPM achieves increased precision at the same computational cost.

We apply and benchmark hybrid KPM by computing supercurrent and Josephson inductance of a long Josephson junction [10–12], where both the contribution of discrete subgap states and the continuum are of the same order. Turning to the effective models, we consider two model systems: tunneling Hamiltonian of two coupled quantum dots [13], and band structure of a semiconductor nanowire [14–17]. In both cases we start from a microscopic Hamiltonian and obtain an accurate effective model, which requires using up to 3rd order perturbation theory.

## II. KERNEL POLYNOMIAL METHOD

To compute thermodynamic properties and effective models, one needs to evaluate the action of the Fermi function or Green's function of the Hamiltonian on a state. The kernel polynomial method (KPM) [9] enables an efficient approximation of such functions of operators. We start by rescaling a Hamiltonian $\hat{H}$ such that its spectrum $\{E_k\}$ is bounded to the interval $(-1, 1)$. In general, a function $f(\hat{H}, \lambda)$ of a Hermitian operator $\hat{H}$ and a set of parameters $\lambda$ can be calculated using the eigendecomposition $\hat{H} = \sum_k E_k |\psi_k\rangle\langle\psi_k|$ as

$$f(\hat{H}, \lambda) \equiv \sum_k f(E_k, \lambda) |\psi_k\rangle\langle\psi_k|, \tag{1}$$

where $f(E, \lambda)$ is a scalar function. The expansion in eigenfunctions is computationally expensive since it requires the full diagonalization of $\hat{H}$. This process scales as $\mathcal{O}(N^3)$ with the $N$ size of the Hilbert space.

* Electronic address: dvarjas@gmail.com

An alternative approach—KPM—utilizes the expansion of the scalar function $f(E, \lambda)$ in terms of Chebyshev polynomials $T_m$

$$f(E, \lambda) = \sum_{m=0}^{\infty} \alpha_m(\lambda) T_m(E), \qquad (2)$$

to build the operator function $f(\hat{H}, \lambda)$ (see Appendix A for expansions of commonly used functions). The Chebyshev polynomials $T_m(x) = \cos(m \arccos x)$ form a complete basis in the interval $(-1, 1)$. They are orthogonal under the inner product

$$\langle f \cdot g \rangle = \int_{-1}^{1} \frac{f(x)g(x)}{\pi\sqrt{1-x^2}} dx, \qquad (3)$$

and satisfy the recursion relation $T_{m+1}(x) = 2xT_m(x) - T_{m-1}(x)$. The Chebyshev coeficients $\alpha_m$ are calculated using the inner product from Eq. (3) with variable $E$

$$\alpha_m(\lambda) = \langle f(E, \lambda) \cdot T_m(E) \rangle, \qquad (4)$$

and the same coefficients apply to the polynomial expansion of the operator function

$$f(\hat{H}, \lambda) = \sum_{m=0}^{\infty} \alpha_m(\lambda) T_m(\hat{H}). \qquad (5)$$

We are interested in the action of $f(\hat{H}, \lambda)$ on a set of vectors. In such situations, the expensive part of the computation is to calculate $T_m(\hat{H}) |v\rangle$, and once we have done that, the coefficients can be readily computed (in most cases analytically) for any value of the parameters $\lambda$.

In practice, the magnitude of the coefficients $\alpha_m$ decays with $m$ and we truncate the series to a finite order $M$. To stabilize the convergence and avoid Gibbs oscillations, while ensuring positivity, we use either the Jackson or Lorentz kernel [9], which is a set of prefactors $g_{m,M}$ that modify the coefficients to $\tilde{\alpha}_m(\lambda) = g_{m,M}\alpha_m(\lambda)$. The recently developed Chebyshev polynomial Green's function method [18] avoids the need for introducing the kernel by approximating a smoothened Green's function. Because we aim to resolve individual states, we do not expect this technique to be useful in the hybrid setting.

The error of the KPM approximation comes from the function $f$ being replaced by its finite order Chebyshev polynomial approximation:

$$f(\hat{H}, \lambda) \overset{\text{KPM}}{\approx} \tilde{f}(\hat{H}, \lambda) \equiv \sum_{k} \tilde{f}(E_k, \lambda) |\psi_k\rangle\langle\psi_k|, \qquad (6)$$

with

$$\tilde{f}(E, \lambda) = \sum_{m=0}^{M} \tilde{\alpha}_m(\lambda) T_m(E). \qquad (7)$$

This error is small if the function is smooth, or there are no eigenvalues of $\hat{H}$ in regions where it changes fast.

The order of the approximation $M$, together with the choice of the kernel, sets the energy resolution of the approximation, which for the Jackson kernel is inversely proportional to $M$. The Chebyshev expansion of order $M$ captures features larger than $W/M$, where $W$ is the full bandwidth of $\hat{H}$ [9].

Hamiltonians and other observables that appear in physical problems are typically sparse matrices where the number of nonzero entries is proportional to the system size $N$. This allows calculating a sparse matrix–vector product in $\mathcal{O}(N)$ time, much faster than the $\mathcal{O}(N^2)$ scaling of dense matrix–vector products. The recursion relation for Chebyshev polynomials can then be rewritten for the operator function acting on a vector as

$$|v_{m+1}\rangle = 2\hat{H} |v_m\rangle - |v_{m-1}\rangle, \qquad (8)$$

where $|v_m\rangle = T_m(\hat{H}) |v\rangle$. Hence, the Chebyshev expanded action $f(\hat{H}, \lambda) |v\rangle$ up to order $M$ can be computed in $\mathcal{O}(NM)$ time. The computational effort of KPM scales as $\mathcal{O}(NW/\Delta)$ where $\Delta$ is the required energy resolution. KPM is most efficient when the desired energy resolution is much coarser than the typical level spacing, that is when $\Delta \gg W/N$, and $M = W/\Delta \ll N$.

The key idea behind hybrid KPM is using a more accurate approximation of $f(\hat{H}, \lambda)$:

$$f(\hat{H}, \lambda) \overset{\text{hybrid}}{\approx} \tilde{f}(\hat{H}, \lambda) - \sum_{k \in A} \tilde{f}(E_k, \lambda) |\psi_k\rangle\langle\psi_k| \\ + \sum_{k \in A} f(E_k, \lambda) |\psi_k\rangle\langle\psi_k|. \qquad (9)$$

Here we combine the KPM approximation of the complete spectrum with the exact contribution of a few states in a small subspace $A$. To avoid double-counting we subtract the KPM contribution of the exactly known states and add back their exact contribution. Approximation (6) has a large error due to states in the energy range where $f$ changes rapidly. Our approach fixes this problem by using a sparse eigensolver to find these states and taking their contribution into account exactly, while keeping the energy resolution of KPM low.

## III. LÖWDIN PERTURBATION THEORY

Quantum systems often have many degrees of freedom, while only a few states (for example the lowest energy ones) are of interest for physical understanding. Perturbative effective models describe such a situation well by constructing a Hamiltonian of the small "interesting" subspace, and integrating out the remaining states. After the integration, the effective model includes both a shift in the energy of the eigenstates and additional coupling terms mixing various eigenstates.

We use the Löwdin partitioning approach [1, 3, 4, 19] (also known as Schrieffer–Wolff transformation [2]) to calculate the effective Hamiltonian. If applied directly, this

approach requires full diagonalization of the unperturbed Hamiltonian, making it unfeasible in large systems. We find, however, that it is sufficient to only exactly know the states in the interesting subspace, and use hybrid KPM to integrate out the remaining states. This allows us to compute effective models in systems with millions of degrees of freedom, as long as the interesting subspace is small.

### A. Löwdin partitioning

We start by separating initial Hamiltonian into unperturbed part $H_0$ and perturbation with $\lambda_\alpha$ as small parameters:

$$H = H_0 + \sum_\alpha \lambda_\alpha H'_\alpha. \tag{10}$$

Assuming that the eigenstates and energies of $H_0$ are known

$$H_0 |\psi_n\rangle = E_n |\psi_n\rangle, \tag{11}$$

we split states $|\psi_n\rangle$ into two groups, $A$ and $B$. We are interested in states from group $A$ whereas the effect of states $B$ we include via perturbation theory. We assume that these two groups of states are separated in energy, but states within $A$ or $B$ may be degenerate. The goal is to find a unitary basis transformation with skew-Hermitian $S$ as

$$\tilde{H} = e^{-S} H e^S, \tag{12}$$

such that the transformed Hamiltonian $\tilde{H}$ does not couple the $A$ and $B$ subspaces, and the block in the $A$ subspace is the effective Hamiltonian, $H_{\text{eff}} = \tilde{H}_{AA}$. We find $S$ and $H_{\text{eff}}$ order-by-order in the small parameters (for details see Appendix B):

$$H_{\text{eff}} = \tilde{H}^{(0)} + \sum_\alpha \lambda_\alpha \tilde{H}^{(1,\alpha)} + \sum_{\alpha\beta} \lambda_\alpha \lambda_\beta \tilde{H}^{(2,\alpha\beta)} + \dots . \tag{13}$$

When the $A$ subspace corresponds to a single eigenvalue, that is possibly degenerate, the Löwdin perturbation theory reproduces the conventional perturbation theory.

### B. The KPM approximation of effective Hamiltonian

To provide a concrete example, we consider the second order effective Hamiltonian with one small parameter,

$$H_{\text{eff}} = \tilde{H}^{(0)} + \lambda \tilde{H}^{(1)} + \lambda^2 \tilde{H}^{(2)}, \tag{14a}$$

with the explicit terms

$$\tilde{H}^{(0)}_{mn} = E_m \delta_{m,n}, \tag{14b}$$

$$\tilde{H}^{(1)}_{mn} = \langle \psi_m | H' | \psi_n \rangle, \tag{14c}$$

$$\tilde{H}^{(2)}_{mn} = \frac{1}{2} \sum_{l \in B} \left( \frac{\langle \psi_m | H' | \psi_l \rangle \langle \psi_l | H' | \psi_n \rangle}{E_m - E_l} \right.$$
$$\left. + \frac{\langle \psi_m | H' | \psi_l \rangle \langle \psi_l | H' | \psi_n \rangle}{E_n - E_l} \right), \tag{14d}$$

where $m$ and $n$ index states of the $A$ subspace and $l$ indexes states of the $B$ subspace.

We rewrite the first term in the second order contribution as

$$\sum_{l \in B} \frac{\langle \psi_m | H' | \psi_l \rangle \langle \psi_l | H' | \psi_n \rangle}{E_m - E_l}$$
$$= \langle \psi_m | H' \left( \sum_{l \in B} \frac{|\psi_l\rangle \langle \psi_l|}{E_m - E_l} \right) H' | \psi_n \rangle$$
$$= \langle \psi_m | H' P_B G_0(E_m) P_B H' | \psi_n \rangle, \tag{15}$$

where $G_0$ is the unperturbed Green's function

$$G_0(E) = \frac{1}{E - H_0} = \sum_i \frac{|\psi_i\rangle \langle \psi_i|}{E - E_i}, \tag{16}$$

and $P_B$ is the projector onto the $B$ subspace.

This formulation is well suited for approximate evaluation using the KPM expanded Green's function. The Green's function only acts on a small set of vectors, $|\phi_n\rangle = P_B H' |\psi_n\rangle$ for $n \in A$. We obtain the exact eigenstates of the $A$ subspace using sparse diagonalization of $H_0$, and compute $P_B$ using $P_B = \mathbb{1} - P_A$. The states $|\phi_n\rangle$ are purely in the $B$ subspace, and we evaluate the Green's function at the energy of a state in the $A$ subspace. The energy separation between the two sets of states removes all divergences, so that the action of $G_0$ is well approximated using KPM with a low energy resolution. After these substitutions, the second order contribution simplifies to

$$\tilde{H}^{(2)}_{mn} = \frac{1}{2} \langle \phi_m | [G_0(E_m) + G_0(E_n)] | \phi_n \rangle. \tag{17}$$

Similar simplification in terms of $G_0$ is also possible for all higher orders, for details see Appendix C.

### C. Effective Hamiltonian with hybrid KPM

In order to accurately approximate the action of $G_0$ on $B$ states closest to the $A$ subspace in energy, we need to choose the number of Chebyshev moments of the order of $W/\Delta$, where $W$ is the full bandwidth of $H_0$ and $\Delta$ is the gap between $A$ and $B$ states. Hence, for small $\Delta$ accurate calculation using KPM becomes computationally expensive. Alternatively, knowing all the $B$ eigenstates

would allow exact evaluation of the Green's function, at even higher computational cost.

To solve this problem, we propose the hybrid KPM approach, where only a subset $B_e$ of the $B$ eigenstates is known explicitly. These we choose to be the eigenstates with closest energy to the $A$ states, and are obtained using sparse diagonalization. We split the Green's function of the $B$ subspace to two terms:

$$G_0(E)P_B = \sum_{l \in B_e} \frac{|\psi_l\rangle \langle \psi_l|}{E - E_l} + G_0^{\text{KPM}}(E)(P_B - P_{B_e}), \quad (18)$$

where $P_B$ and $P_{B_e}$ are projectors to the $B$ and $B_e$ subspaces, and $G_0^{\text{KPM}}$ is the KPM approximated Green's function.

## IV. COMPUTATION OF THERMODYNAMIC QUANTITIES

### A. Evaluation of operator expectation values

Physical observables in a non-interacting fermionic system are thermal expectation values of a Hermitian operator $\hat{A}$:

$$\langle \hat{A} \rangle_{E_F} = \sum_k f(E_k, E_F) \langle \psi_k| \hat{A} |\psi_k\rangle, \quad (19)$$

where the sum runs over all eigenstates of the Hamiltonian $|\psi_k\rangle$ with eigenenergies $E_k$. The occupation of the states is given by the Fermi function

$$f(E, E_F) = \frac{1}{e^{\beta(E - E_F)} + 1} \quad (20)$$

with $\beta = (k_B T)^{-1}$ and $E_F$ the Fermi energy. Converting the sum to an integral over energy by inserting a delta function, we introduce the spectral density of the operator $A(E) \equiv \text{Tr}\left[\hat{A}\, \delta(E - \hat{H})\right]$, yielding

$$\langle \hat{A} \rangle_{E_F} = \int dE\, f(E, E_F) A(E). \quad (21)$$

This can be rewritten as a trace using the operator function formalism, and readily evaluated using KPM:

$$\langle \hat{A} \rangle_{E_F} = \text{Tr}\left[\hat{A}\, f(\hat{H}, E_F)\right] = \sum_m \tilde{\alpha}_m(E_F)\mu_m, \quad (22)$$

where the KPM moments are

$$\mu_m = \text{Tr}\left[\hat{A}\, T_m(\hat{H})\right]. \quad (23)$$

The Fermi function changes rapidly in the interval $1/\beta$ around the Fermi level. Our strategy is to compute the states near the Fermi level exactly and approximate the rest of the states using low order KPM. Following the

hybrid KPM approximation, we substitute Eq. (9) into Eq. (22):

$$\langle \hat{A} \rangle_{E_F} \approx \sum_m \tilde{\alpha}_m(E_F)\left(\mu_m - \mu_m^A\right)$$
$$+ \sum_{i \in A} f(E_i, E_F) \langle \psi_i| \hat{A} |\psi_i\rangle, \quad (24)$$

where the KPM moments restricted to the $A$ subspace are

$$\mu_m^A = \sum_{i \in A} T_m(E_i) \langle \psi_i| \hat{A} |\psi_i\rangle. \quad (25)$$

The trace in the full contribution is efficiently approximated using the stochastic trace approximation [9]. The exact evaluation of the trace is also feasible if the operator $\hat{A}$ has low rank and the basis of its image space is explicitly known:

$$\mu_m = \sum_{|\psi\rangle \in \text{Im}\, \hat{A}} \langle \psi| \hat{A}\, T_m(\hat{H}) |\psi\rangle. \quad (26)$$

### B. Perturbative KPM

We now generalize KPM to allow order-by-order expansion of thermodynamic expectation values. We consider a generic function $g$ and perturbed Hamiltonian $\hat{H} = \hat{H}_0 + \lambda\hat{H}_1$ where $\lambda$ is a small parameter. Our goal is to evaluate $\text{Tr}\left[g(\hat{H})\right]$ order-by-order in the small parameter. For example, the expectation value of the energy of a filled Fermi sea is $\langle E \rangle = \text{Tr}\left[g(\hat{H})\right]$ with $g(E) = Ef(E, E_F)$. Our method also applies to expressions of the form $\text{Tr}\left[\hat{A}g(\hat{H})\right]$, but we restrict to the $\hat{A} = \mathbb{1}$ case for brevity.

Our idea is to keep track of parameter dependence when computing the Chebyshev recursion relation (8), allowing $\hat{H}$ and $|v_m\rangle$ to be polynomials of $\lambda$. Since we are only interested in the result up to $\lambda^n$ order, we discard all higher order terms at every step of the iteration, resulting in KPM moments $\mu_m$ and expectation value $\text{Tr}\left[g(\hat{H})\right]$ that is also an $n$'th order polynomial of $\lambda$. At a finite number of moments $M$ this method reproduces the series expansion of $\text{Tr}\left[\tilde{g}(\hat{H})\right]$ in $\lambda$, where $\tilde{g}$ is the Chebyshev approximation of $g$. The resulting increase in computational cost scales as $\mathcal{O}(n^2)$, making this method feasible at low expansion orders.

Rearranging the terms in the perturbation expansion allows us to efficiently calculate the trace when the image space of the perturbation $H_1$ is small. After a cyclic permutation inside the trace, the $\lambda$-linear term in the expansion becomes

$$\frac{d}{d\lambda} \text{Tr}\left[g(\hat{H})\right]_{\lambda=0} = \text{Tr}\left[g'(\hat{H}_0)\hat{H}_1\right], \quad (27)$$

where $g'$ is the derivative of $g$. Because $\hat{H}_1$ is the rightmost operator, the trace reduces to a sum over a basis of the image space of $\hat{H}_1$. Applying this to the energy expectation value in the zero temperature limit where $f$ is a step function [using that $x\delta(x) = 0$] we get

$$\frac{d}{d\lambda}\langle E\rangle_{\lambda=0} = \frac{d}{d\lambda}\operatorname{Tr}\left[\hat{H}f(\hat{H})\right]_{\lambda=0}$$
$$= \operatorname{Tr}\left[f(\hat{H}_0)\hat{H}_1\right] = \langle\hat{H}_1\rangle_{\lambda=0} \qquad (28)$$

which is the same as the ground state expectation value of $H_1$, and was already discussed in the previous section. Similar simplifications apply to all higher orders; here we present the second order case in detail. For a generic function $g$ and Hamiltonian $\hat{H} = \hat{H}_0 + \lambda\hat{H}_1 + \lambda^2\hat{H}_2$, by expanding and permuting terms proportional to $\lambda^2$ we find

$$\frac{1}{2}\frac{d^2}{d\lambda^2}\operatorname{Tr}\left[g(\hat{H})\right]_{\lambda=0}$$
$$= \operatorname{Tr}\left[g'(\hat{H}_0)\hat{H}_2\right] + \frac{1}{2}\frac{d}{d\lambda}\operatorname{Tr}\left[g'(\hat{H})\hat{H}_1\right]_{\lambda=0} \qquad (29)$$

To evaluate the traces we sum over the basis of the image spaces of $\hat{H}_2$ and $\hat{H}_1$ respectively in the two terms. To obtain the second term we use the KPM expansion to first order in $\lambda$.

To apply the hybrid KPM approach to the perturbative KPM, we utilize the hybrid KPM Löwdin perturbation theory developed in Sec. III, obtaining the perturbation series of the $A$ subspace eigenenergies. We treat a single eigenpair $(E_k, |\psi_k\rangle)$ of $\hat{H}_0$ as the $A$ subspace for the purposes of Löwdin perturbation theory, and use the rest of the exactly known states as the $B_e$ subspace in the hybrid evaluation of the Green's function. Repeating this for every $A$ eigenstate produces the power series expansions of the perturbed $E_k$ up to the desired order. Combining the perturbative KPM result for the full spectrum and substituting the Löwdin expansion of the $A$ subspace we obtain

$$\operatorname{Tr}\left[g(\hat{H})\right] \stackrel{\text{hybrid}}{\approx} \operatorname{Tr}\left[\tilde{g}(\hat{H})\right] - \sum_{k\in A}\tilde{g}(E_k) + \sum_{k\in A}g(E_k). \quad (30)$$

We compute the series expansion of the last two terms in $\lambda$ using the Taylor series of $g$ and $\tilde{g}$. The hybrid Löwdin approximation has the highest accuracy for states in the middle of the $A$ subspace energy range, which we choose to coincide with the fastest-changing region of $g'$. For states close to the edge of the $A$ subspace energy range the approximation is less accurate, but, at the same time, the difference between $g$ and $\tilde{g}$ is also small, resulting in a small overall error.

## V. APPLICATIONS

### A. Supercurrent and Josephson inductance

As an illustration we apply the hybrid KPM method to calculate supercurrent in a Josephson junction [10]. When the Thouless energy is smaller than the superconducting gap $\Delta$—the so-called long-junction regime [11, 12]—the continuum spectrum at $|E| > \Delta$ also responds to the superconducting phase difference and contributes to the total supercurrent [20]. In the hybrid KPM approach we calculate the subgap states using exact diagonalization, and estimate the contribution of continuum states using KPM.

We consider a Josephson junction with a normal region of length $L_N$, superconducting leads of length $L_S$ and width $W$ as shown in the inset of Fig. 1. For simplicity, we consider a spinless Bogoliubov-de Gennes (BdG) Hamiltonian without magnetic field:

$$H_{BdG} = \begin{pmatrix} \frac{\mathbf{p}^2}{2m} - \mu & \Delta(x) \\ \Delta^*(x) & \mu - \frac{\mathbf{p}^2}{2m} \end{pmatrix}, \qquad (31)$$

with $\mathbf{p}$ the momentum operator, $m$ the effective electron mass, and $\mu$ the chemical potential. The superconducting order parameter $\Delta(x)$ is zero in the normal region and $\Delta$ in the superconducting leads. We discretize this Hamiltonian on a square lattice with lattice constant $a$ and a nearest neighbour hopping $t = \hbar^2/(2ma^2)$. We introduce the superconducting phase difference $\phi$ through a Peierls substitution

$$H_{ij} \rightarrow \begin{cases} H_{ij}\exp(i\phi\tau_z/2) & \text{if } i \in L \text{ and } j \in R \\ \exp(-i\phi\tau_z/2)H_{ij} & \text{if } i \in R \text{ and } j \in L, \quad (32) \\ H_{ij} & \text{otherwise} \end{cases}$$

with $H_{ij}$ the hopping Hamiltonian between site $i$ and $j$ in the BdG formalism, $\tau_z$ the Pauli matrix in particle-hole space. Finally, $L$ and $R$ correspond to the left and right sides of a cut in the normal region parallel to the normal-metal–superconductor interface (see the inset of Fig. 1). The current operator across the cut is the derivative of the Hamiltonian with respect to the flux:

$$\hat{I} = \frac{2e}{\hbar}\frac{d\hat{H}}{d\phi}. \qquad (33)$$

In order to calculate the KPM contribution to the trace

$$\langle\hat{I}\rangle = \operatorname{Tr}\left[\hat{I}f(\hat{H})\right], \qquad (34)$$

we use the basis of the sites next to the cut. All other states are annihilated by the current operator and do not contribute to the current.

Here and in the rest of the manuscript we use the Kwant software package [21] to construct tight-binding Hamiltonians. We consider a Josephson junction of length $L_N = L_S = 50a$ and width $W = 15a$ and set the parameters $\mu = 0.2t$ and $\Delta = 0.15t$. As explained in sec. IV, we calculate the subgap Andreev bound states exactly using sparse diagonalization and treat them as the $A$ subspace. We show the spectral density of the current operator $I(E)$ in Fig. 1, where we plot the contributions of subgap and continuum states separately. The KPM spectrum of the

current operator vanishes at this resolution with $M = 500$ moments. The contribution of only the continuum states calculated with hybrid KPM is, however, non-vanishing, and the exactly known subgap states contribute Dirac delta peaks.

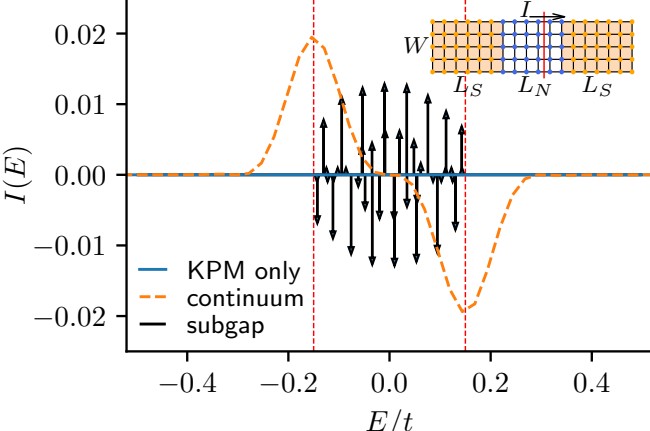

FIG. 1. Current operator spectrum as a function of energy with fixed relative superconducting phase of $\pi/2$. The solid blue line represents the KPM only spectrum of the current operator; the arrows represent the Dirac delta contributions of subgap states; the dashed orange line shows the contribution of the continuum states. Inset: Sketch of the system. The shaded regions are superconducting with a normal region in the middle. The red line represents the cut for which we calculate the supercurrent.

We compute the contributions of the Andreev states and of the continuum states to the current-phase relation, with the result shown in Fig. 2. The contributions of both the subgap and the continuum states are significant, while their sum agrees with the exact result to a high precision.

To demonstrate hybrid KPM in higher order perturbation theory, we turn to the Josephson inductance. The inverse of the junction inductance is equal to the derivative of the current expectation value with respect to the flux:

$$L_J^{-1} = \frac{(2e)^2}{\hbar^2} \frac{d^2}{d\phi^2} \langle \hat{H} \rangle. \tag{35}$$

We evaluate this expression using the hybrid method discussed in Sec. IV B taking into account the second derivative of the Hamiltonian, with the result shown in Fig. 3. The sharp peak in $L_J^{-1}$ at $\phi = \pi$ is accurately captured by the direct evaluation of the second derivative using our method, while accurate calculation using a discrete derivative of the current expectation value requires a much higher resolution in $\phi$. As we observed in Sec. IV B, the hybrid Löwdin perturbation theory estimates the energies of the states near the edge of the $A$ subspace with a low precision. This is why the contribution of the bound states to $L_J^{-1}$ disagrees with the derivative of the bound state contribution to the current shown in Fig. 2. Nevertheless, because this error cancels with the $B$ subspace

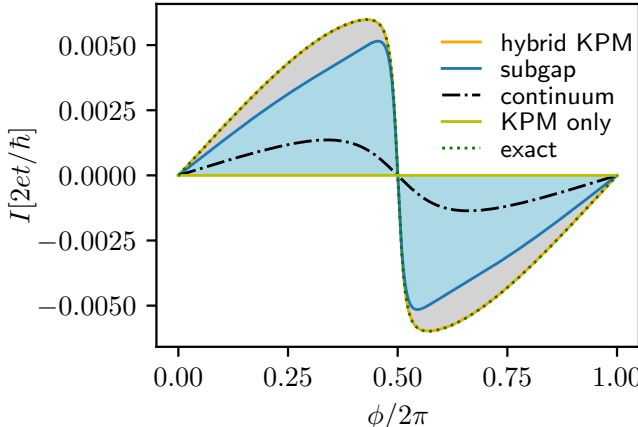

FIG. 2. Supercurrent as a function of the superconducting phase difference. The orange line is the total supercurrent through a Josephson junction calculated with hybrid KPM, whereas the blue and black lines are the contributions from subgap and continuum states respectively. The hybrid KPM result agrees with full diagonalization, while the pure KPM estimate vanishes.

contribution, the precision of the full result remains the same.

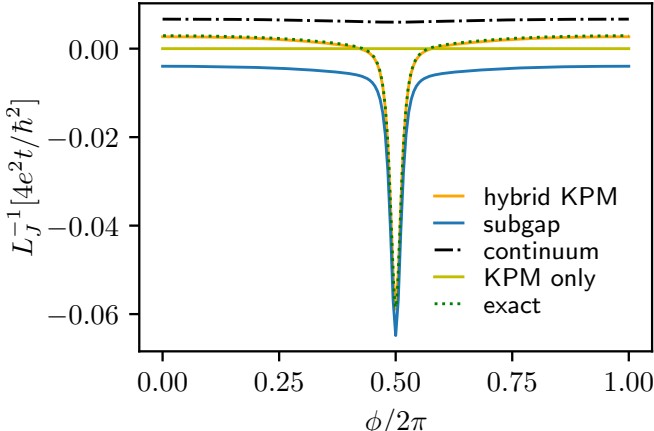

FIG. 3. Inverse Josephson inductance as a function of the superconducting phase difference. The orange line represents $L_J^{-1}$ calculated with hybrid KPM, whereas the blue and black lines show the corresponding contributions from subgap and continuum states respectively. The hybrid KPM result agrees with the exact result using full diagonalization, while the pure KPM result vanishes.

The zero temperature limit is the most computationally expensive both to pure KPM and imaginary energy integration [22, 23]. Computing the finite temperature results within hybrid KPM, however, amounts to replacing $f$ with the Fermi function at the correct temperature. Because the computational cost of hybrid KPM is dominated by the computation of the KPM moments and the perturbation expansion of low-lying states, the extra computational cost of a temperature sweep is negligible.

## B. Effective double quantum dot Hamiltonian

Turning to the hybrid Löwdin perturbation theory, we apply hybrid KPM to calculate an effective Hamiltonian of several low energy states in a double quantum dot system. In order to use the basis of individual quantum dot states, we start with a system with decoupled dots and include hoppings between the dots perturbatively. When the tunnel barrier between the dots is low, the eigenstates become strongly hybridized, so that the perturbation theory requires a sufficiently high order in the inter-dot coupling. We address this need by including the eigenfunctions and eigenenergies of the lowest few bound states exactly and treating the remaining part of the energy spectrum up to third order in the Löwdin perturbation theory using hybrid KPM approach.

We consider two gate-defined quantum dots formed in a quantum well with the interdot tunnel coupling and dot chemical potential controlled by the gate electrodes. In the continuum approximation the quantum well Hamiltonian is

$$H_{2D} = \frac{\hbar^2}{2m_e}(k_x^2 + k_y^2) + V(x, y), \qquad (36)$$

where $m_e$ is the effective electron mass, $k_x$ and $k_y$ are the components of the electron wave vector, and $V(x, y)$ is the electrostatic potential. We discretize the continuum Hamiltonian $H_{2D}$ using the finite difference approximation on a square lattice with a lattice constant of $5\,\mathrm{nm}$ and the effective mass of GaAs. We consider the gate geometry of Ref. 13, with the gate electrodes $60\,\mathrm{nm}$ above the quantum well. Plunger gates control the dot chemical potential, while the tunnel barrier height between the dots is controlled by the barrier gate in the middle, as shown in Fig. 4. We calculate the electrostatic potential induced in the quantum well using the approximation of Ref. [24]. In the initial configuration the gate potentials form two tunnel-coupled quantum dots, as shown in Fig. 4.

We separate the Hamiltonian into a sum of unperturbed $H_0$ term and two perturbation terms:

$$H = H_0 + \lambda_g \Delta H_g + \lambda_c H_c. \qquad (37)$$

Here $H_0$ is the initial Hamiltonian with the hoppings between the left and the right halves of the system removed, $\Delta H_g$ is the deviation of the gate potential from the initial setting, and $H_c$ is the hoppings connecting the left and right halves. We split the spectrum of the Hamiltonian into two subspaces: $A$ contains the two lowest bound states in each quantum dot, and $B$ the rest of the energy spectrum. We obtain the states in $A$ and a few states in $B_e \subset B$ with the lowest energies using sparse diagonalization. Setting $\lambda_c = 1$ reproduces the original Hamiltonian without the cut between the dots. We select the perturbation $\Delta H_g$ as the potential resulting from an antisymmetric detuning of the plunger gates by $\pm \Delta V/2$. In Fig. 5, we compare the eigenenergies calculated from the effective model using first and third order perturbation theory with the sparse diagonalization results. The

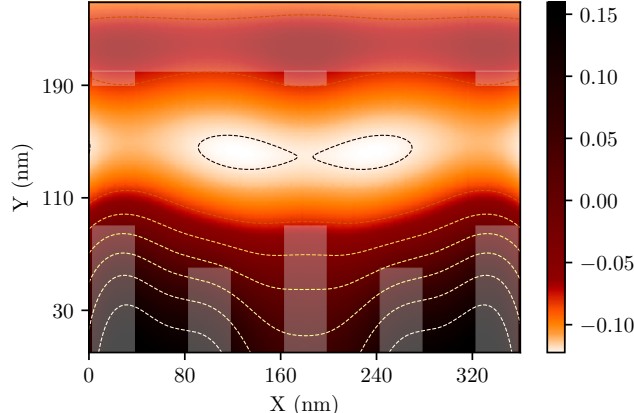

FIG. 4. The 2DEG electrostatic potential superimposed with gate electrodes deposited on top of the GaAs heterostructure. We use the gate design and heterostructure from [13]. Plunger and barrier gates are shown at the bottom and screening gates at the top. By applying negative voltage to the gate electrodes, we locally deplete the 2DEG to form two quantum dots as represented by the equipotential lines.

first order perturbation does not require hybrid KPM and coincides with conventional first order perturbation theory. It cannot, however, accurately estimate the spectrum, while third order Löwdin perturbation theory using hybrid KPM shows a good agreement with the exact result.

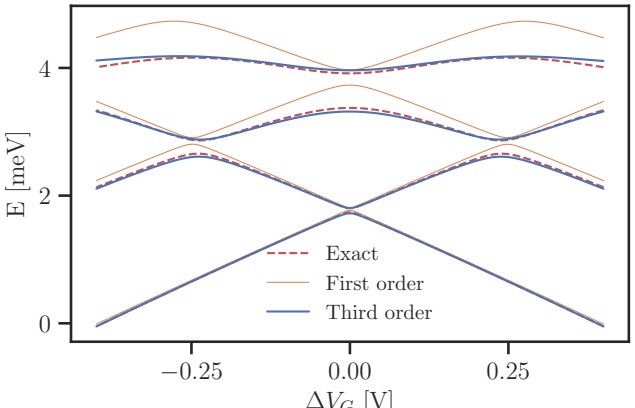

FIG. 5. Energy spectrum as a function of gate voltage difference between the two dots. Eigenenergies calculated from the first and third order effective models are compared against the exact energies.

## C. Effective band structures

Semiconductor nanowires, besides many potential applications [14], are of interest as a platform to realize Majorana states when proximitized with a superconduc-

tor [15–17]. The necessary ingredients for the creation of Majorana states are spin-orbit interaction and external magnetic field, which remove spin degeneracy in the lowest subband of the wire, resulting in effective $p$-wave superconducting pairing. In an external electric field normal to the wire the bulk spin-orbit coupling of the semiconductor results in Rashba spin-orbit interaction. The minimal model describing the relevent phenomena, with the exception of superconductivity, is the 2-band effective model:

$$H = \frac{\hbar^2}{2m^*}k_z^2 + \mu + \alpha k_z\left(\sigma_y E_x + \sigma_x E_y\right) + \mu_B \mathbf{B}g\boldsymbol{\sigma}, \quad (38)$$

where $m^*$ is the effective mass of the lowest subband, $k_z$ is the momentum along the wire, $\mu$ is the chemical potential, $E_x$ and $E_y$ are components of the electric field, $\boldsymbol{\sigma}$ are the Pauli matrices and $\alpha$ is the strength of the Rashba spin-orbit interaction. The external magnetic field is $\mathbf{B}$, $\mu_B$ is the Bohr-magneton and $g$ is the effective $g$-factor tensor in the lowest subband. While this simple model is easy to solve, extracting the parameters of realistic setups starting from a microscopic model is computationally hard. We solve this task using hybrid KPM.

We start from the 8-band $\mathbf{k} \cdot \mathbf{p}$ model of bulk zinc-blende materials, in particular InAs [4, 19, 25, 26]. This continuum model accurately captures the $s$-type conduction and $p$-type valence bands near the Fermi level at small momenta, up to second order in $k$. We consider an infinite wire oriented along the $z$-axis with approximately circular cross-section in the $xy$ plane with radius $R$. We discretize the $\mathbf{k} \cdot \mathbf{p}$ Hamiltonian in the $xy$-plane by replacing momenta $k_x$ and $k_y$ (but not $k_z$) with discrete spatial derivatives. We include the Zeeman term with the bulk $g$-factor $g^* = -15$ of InAs [4, 27]. We introduce the orbital magnetic field using Peierls substitution, as well as the electrostatic potential $V = -E_x x - E_y y$. For this example we use radius $R = 25$ nm with discretization grid lattice constant $a = 1$ nm. This results in a tight-binding model with 31056 degrees of freedom, outside of the practical limits of full diagonalization on a single computer.

We use the Löwdin algorithm treating the tight binding Hamiltonian with vanishing external fields and $k_z = 0$ as the unperturbed Hamiltonian, and include perturbations up to second order in $k_z$ and the electric field, and up to linear order in the magnetic field. Using second order perturbation theory, we obtain the effective model of the form (38) with $m^* = 0.023m_0$, $\alpha = 2.67$ nm$^2$, $\mu = 0.43$ eV, $g_{xx} = g_{yy} = -15.4$ and $g_{zz} = -15.5$, with $m_0$ the free electron mass and all other terms approximately vanishing. This perturbative treatment, only accurate at small parameter values, does not capture the overall energy shift of the subbands resulting from the electrostatic field at field strengths relevant to experiments. Hence, we also construct the effective model using $E_{x0} = 10$ meV/nm as the unperturbed Hamiltonian. The resulting spectrum at finite $k_z$ and $B_z$ agrees with the exact eigenenergies of the full model as illustrated in Fig. 6.

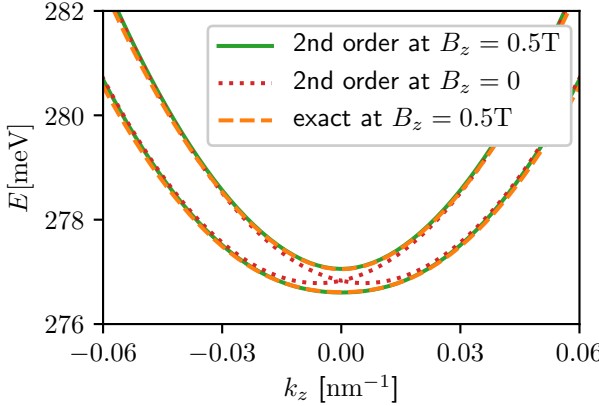

FIG. 6. Energy spectrum of the lowest subband of an InAs nanowire of radius 5 nm. The plot shows the exact result from sparse diagonalization with $B_z = 0.5$ T and $E_x = 10$ meV/nm, and the perturbation theory result around $E_{x0} = 10$ meV/nm at 2nd order at $B_z = 0$ and $B_z = 0.5$ T.

## VI. CONCLUSIONS

We developed the hybrid kernel polynomial method, where we combine the strengths of KPM in treating many states at a low energy resolution and of sparse diagonalization in treating few states with high accuracy. We applied this method to the problems of accurate calculation of expectation values, perturbation theory of thermodynamic quantities and construction of perturbative effective models. The source code of these general and reusable algorithms is available at [28], together with the source code and data of the examples showcased in the manuscript.

We applied our method to several active research topics in condensed matter and mesoscopic physics: calculation of supercurrent and inductance in a Josephson junction, design of spin qubits defined in a two dimensional electron gas, and the calculation of the effective band structure in a realistic model of a semiconductor nanowire. Our examples illustrate how the combination of low and high resolutions enables the investigation of response functions and effective models in systems whose size would make this prohibitively expensive using other approaches.

We did not yet address the following relevant questions:

- What is the optimal way to choose the number of the exactly calculated states and the number of moments in hybrid KPM to minimize the computational effort required for given precision?

- How quickly does the stochastic trace approximation converge in the perturbative KPM scheme?

- What is the general form of rearranged equations similar to (29) for higher orders and multiple perturbation parameters?

- How does the efficiency of our method compare to other recursive numerical approaches to perturbation theory, such as Ref. 29?

These we leave to future work.

Because our method allows treatment of Hilbert spaces up to millions of degrees of freedom, we expect it to be useful in treating interacting quantum mechanical problems. We conjecture that the hybrid approach will also improve KPM-assisted self-consistent mean-field [30] and density matrix renormalization group [31, 32] calculations. Accurate simulation of nanoelectronic devices with truncated few-electron Hilbert spaces is also a promising future direction of research using this methodology.

## ACKNOWLEDGMENTS

We are thankful to T. Rosdahl and J. B. Weston for their role in the development of Qsymm [33], the data structures of which the implementation of our algorithms relies on. This work was supported by ERC Starting Grant 638760, the Netherlands Organisation for Scientific Research (NWO/OCW) as part of the Frontiers of Nanoscience program, NWO VIDI grant 680-47-53, the US Office of Naval Research, and the European Union's Horizon 2020 research and innovation programme under grant agreement No 824140.

## AUTHOR CONTRIBUTIONS

A. R. Akhmerov and P. M. Perez-Piskunow proposed the idea of hybrid KPM and initiated the project. P. M. Perez-Piskunow implemented the conventional KPM algorithms. M. Wimmer proposed and R. Skolasinski implemented the automated Löwdin perturbation theory using full diagonalization. M. Wimmer proposed using KPM for Löwdin perturbation theory, P. M. Perez-Piskunow and D. Varjas implemented hybrid Löwdin perturbation theory. S. R. Kuppuswamy applied Löwdin perturbation theory to the spin qubit example. D. Varjas and R. Skolasinski applied Löwdin perturbation theory to the nanowire example. P. M. Perez-Piskunow and M. Irfan implemented the hybrid expectation value calculation. D. Varjas proposed and implemented the hybrid expectation value perturbation theory. M. Irfan applied these methods to the Josephson junction example. A. R. Akhmerov and M. Wimmer oversaw the project and the code development. All authors took part in writing the manuscript.

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

## Appendix A: Chebyshev polynomial expansion of selected functions

We explicitly give the expansion of a few common functions used in condensed-matter physics and this manuscript: Dirac delta function (used in spectral densities):

$$\delta\left(E - \hat{H}\right) = \frac{1}{\pi\sqrt{1-E^2}} \sum_m \frac{2}{1+\delta_{m,0}} T_m(E) T_m(\hat{H}). \tag{A1}$$

The Green's functions:

$$G^{\pm}(E, \hat{H}) = \lim_{\eta \to 0^+} \frac{1}{E - \hat{H} \pm \eta i} = \mp \frac{2i}{\sqrt{1-E^2}} \sum_m \frac{1}{1+\delta_{m,0}} \exp\left(\pm i\,m \arccos(E)\right) T_m(\hat{H}). \tag{A2}$$

## Appendix B: Details of Löwdin expansion

We adapt this section from Refs. 19 and 34, that closely follows the derivation of arbitrary order quasi-degenerate perturbation theory in Ref. 4. The goal is to find a unitary basis transformation (Schrieffer–Wolff transformation) with skew-Hermitian matrix $S$ ($S^\dagger = -S$) as

$$\tilde{H} = e^{-S} H\, e^S, \tag{B1}$$

such that the transformed Hamiltonian $\tilde{H}$ does not couple the $A$ and $B$ subspaces. The transformation should be the identity when the perturbation vanishes and we expand $S$ as a series in successive orders of the perturbation

$$S = \sum_{j=1}^{\infty} \lambda^j S^{(j)}. \tag{B2}$$

The transformed Hamiltonian (using the Baker-Campbell-Hausdorff formula) is

$$\tilde{H} = \sum_{j=0}^{\infty} \frac{1}{j!} [H, S]^{(j)} = \sum_{j=0}^{\infty} \frac{1}{j!} [H_0 + \lambda H'_d, S]^{(j)} + \sum_{j=0}^{\infty} \frac{1}{j!} [\lambda H'_n, S]^{(j)}, \tag{B3}$$

where the nested commutator $[A, B]^{(j)}$ is defined as

$$[A, B]^{(j)} = [\ldots [[A, \underbrace{B], B], \ldots, B}_{j \text{ times}}], \tag{B4}$$

with commutator $[A, B] = AB - BA$ and we split the perturbation into block-diagonal and block off-diagonal parts as $H' = H'_d + H'_n$ with with $(H'_d)_{AB} = (H'_d)_{BA} = (H'_n)_{AA} = (H'_n)_{BB} = 0$ ($X_{AB}$ denotes the restriction of operator $X$ to the $AB$ block). The requirement on the $S$ we seek is $\tilde{H}_{AB} = \tilde{H}_{BA} = 0$ and we call $\tilde{H}_{AA}$ the effective Hamiltonian. We choose $S$ to be block off-diagonal such that $S_{AA} = S_{BB} = 0$, this removes arbitrary unitary transformations within the $A$ and $B$ subspaces from the result.

To do $n$'th order perturbation theory we demand the equations to be satisfied for all terms up to $\lambda^n$. Separating terms that contribute to diagonal and off-diagonal terms ($\tilde{H} = \tilde{H}_d + \tilde{H}_n$ with $(\tilde{H}_d)_{AB} = (\tilde{H}_d)_{BA} = (\tilde{H}_n)_{AA} = (\tilde{H}_n)_{BB} = 0$) we find:

$$\tilde{H}_d = \sum_{j=0}^{\infty} \frac{1}{(2j)!}[H_0 + \lambda H'_d, S]^{(2j)} + \sum_{j=0}^{\infty} \frac{1}{(2j+1)!}[\lambda H'_n, S]^{(2j+1)}, \tag{B5a}$$

$$\tilde{H}_n = \sum_{j=0}^{\infty} \frac{1}{(2j+1)!}[H_0 + \lambda H'_d, S]^{(2j+1)} + \sum_{j=0}^{\infty} \frac{1}{(2j)!}[\lambda H'_n, S]^{(2j)}. \tag{B5b}$$

Our goal is to recursively find $S^{(n)}$ form the lower orders $S^{(j)}$ for $j \in [1 \ldots n-1]$. We solve $\tilde{H}_n = 0$ up to $n$'th order by inserting the expansion $S = \sum_{j=1}^{n} \lambda^j S^{(j)}$ into (B5b) and letting the sums in $j$ run to $\lfloor (n-1)/2 \rfloor$, this produces all terms up to $n$'th order. We observe that at $n$'th order $S^{(n)}$ only appears in a single commutator, allowing to rearrange the $n$'th order terms in the equation $\tilde{H}_n = 0$ as

$$[H_0, S^{(n)}] = Y^{(n)} \tag{B6}$$

where $Y^{(n)}$ only depends on lower orders of $S$. We generate the $Y$'s using symbolic computer algebra. The first few terms are:

$$[H_0, S^{(1)}] = Y^{(1)} = -H'_n, \tag{B7a}$$

$$[H_0, S^{(2)}] = Y^{(2)} = -[H'_d, S^{(1)}], \tag{B7b}$$

$$[H_0, S^{(3)}] = Y^{(3)} = -[H'_d, S^{(2)}] - \frac{1}{3}[[H'_n, S^{(1)}], S^{(1)}]. \tag{B7c}$$

As the $Y$'s are purely off-diagonal Hermitian, it is possible to write only $Y^{(n)}_{AB}$ in terms of $S_{AB}$, $S_{BA}$ and the restricted components of $H$.

The equations (B7) can be iteratively solved as

$$S^{(j)}_{ml} = \frac{Y^{(j)}_{ml}}{E_m - E_l} \tag{B8}$$

where indices $m$ and $l$ correspond to states in the $A$ and $B$ subspace respectively. With the $n-1$ order expansion of $S$ at hand, we substitute it into (B5a) with the sum over $j$ running to $\lfloor n/2 \rfloor$, or directly into (B3) with the sum over $j$ running to $n$, to produce $\tilde{H}_d$ up to $n$'th order.

The same algorithm works in the case of multiple expansion parameters by replacing $\lambda H'$ with $\sum_\alpha \lambda_\alpha H'_\alpha$ and only keeping track of terms with total power $j$ in the $\lambda_\alpha$ in $S^{(j)}$ and $Y^{(j)}$. Finally, we write the $AA$ block of the transformed Hamiltonian as a sum of successive orders of the perturbation to obtain the effective Hamiltonian:

$$H_{\text{eff}} = \tilde{H}^{(0)} + \sum_\alpha \lambda_\alpha \tilde{H}^{(1,\alpha)} + \sum_{\alpha\beta} \lambda_\alpha \lambda_\beta \tilde{H}^{(2,\alpha\beta)} + \ldots. \tag{B9}$$

## Appendix C: Using KPM in higher order Löwdin expansion

To use KPM efficiently, we want to avoid using an explicit basis for the $B$ subspace. We observe that the expressions (B7) for $Y$ and (B5a) for $\tilde{H}_{AA}$ can be expanded in terms of the restricted operators (i.e. $H'_{AA}$, $H'_{AB}$, etc.). Whenever two terms with $A$ indices are adjacent, we may insert a projector onto the $A$ states $P_A = \sum_m |m\rangle\langle m|$ with a full basis of $A$ states $|m\rangle$. Whenever two terms with $B$ indices are adjacent, we insert a projector onto the $B$ subspace $P_B = \mathbb{1} - \sum_m |m\rangle\langle m|$. This allows to remove the restriction from one of the adjacent terms, for example

$$\langle m|S_{AB}H'_{BB}S_{BA}|m'\rangle = \langle m|P_A S P_B P_B H' P_B P_B S P_A|m'\rangle = \langle m|SH'S|m'\rangle = \sum_{ij} S_{mi}H'_{ij}S_{jm'} \tag{C1}$$

where we used that $S$ is only nonzero in the off-diagonal blocks. This allows to only store the mixed matrix elements $S_{mi} = \langle m | S | i \rangle$ where $|i\rangle$ is the original basis where the Hamiltonian is sparse with indices $i$, $j$ running over the full Hilbert space, and $|m\rangle$ is the basis of the $A$ subspace. In this basis $\sum_i S_{mi} (P_B)_{ij} = S_{mj}$, similarly for block off-diagonal matrices. It is possible to replace all $H_{BB}$ terms with $H$ because there is only one $H$ in every product, all the other terms are $S$'s. This is advantageous as $H'$ acting on the full Hilbert space of size $N$ can be represented as a sparse matrix of $\mathcal{O}(N)$ nonzero entries, while $S_{mi}$ and other off-diagonal components can be stored as small dense matrices with $\mathcal{O}(Na)$ entries where $a = \dim(A)$.

Now we rewrite (B8) in terms of the Green's function:

$$S_{mi}^{(n)} = \sum_j Y_{mj}^{(n)} \left( \frac{1}{E_m - H_0} \right)_{ji} = \sum_j \left[ G_0(E_m)_{ij} \left( Y^{(n)\dagger} \right)_{jm} \right]^\dagger \tag{C2}$$

where we used that $Y$ is block off-diagonal and $G_0(E)$ does not mix the $A$ and $B$ subspaces. For numerical stability reasons, we still apply $P_B$ from the right in practice. Following the procedure outlined in Appendix B we successively generate all $S$ terms and produce $\tilde{H}_{AA}$, the only difference is using the above basis convention.

The computational complexity of generating the $n$'th order effective Hamiltonian (in the case of a single small parameter) is $\mathcal{O}(n^2 a N M)$, where $M$ is the number of KPM moments, practically chosen to be at the order of bandwidth/gap. We obtain this estimate by the following reasoning: A single evaluation of the KPM Green's function on a vector costs $\mathcal{O}(NM)$. To get $S^{(j)}$, we need to apply $G$ to $(aj)$ vectors on the right hand side, as $Y^{(j)}$ is a $j$'th order polynomial of the small parameter. We argue that the KPM step is the costliest part of the procedure, because evaluation of $Y$ and $\tilde{H}$ only involves products of small or sparse matrices.

There is, however, a combinatorial factor in the number of terms involved in these expressions, which grows exponentially with $j$. At high orders $Y^{(n)}$ contains $\mathcal{O}(2^n)$ terms with a single small parameter. At high enough orders, it is more efficient to directly evaluate the commutator series giving $Y^{(n)}$ by substituting the $n - 1$ order expansion of $S$ with numerical coefficients. Truncating to terms of at most order $n$ after every multiplication, this only takes $\mathcal{O}(n^3)$ time. Hence, this latter method becomes more efficient for high enough orders. Combinatorial factors are even larger if there are multiple small parameters in the expansion. We defer further analysis of the complexity and possible optimizations of high order expansions.