# Peer review of "Hybrid kernel polynomial method"

_SciPost Physics_

## Round 2 · Referee Report · Anonymous (Referee 1) · 2019-12-27

Strengths

to deal with finite number of discrete localized states in hybrid kernel polynomial method (Chebyshev approach)

Weaknesses

discussion lacks clarity

Report

The paper aims to improve the Cheyshev approach to deal with large systems still within the range of exact diagonalization (ED) yet in the presence of discrete states. The paper is valid and deserves publication eventually. However, the paper lacks clarity and rigor which needs to be improved prior to publication.

When introducing the key idea behind hybrid KPM with Eq. 9, the author states that they include the "exact contribution of a few states in a small "interesting" subspace A." This is too general: I do not believe that it is beneficial to include the exact solution of a small interesting subspace in general. It appears what the authors really mean throughout the paper, is the presence of a finite number of discrete localized states, e.g., by spatial confinement or ingap states. This needs to be clearly emphasize throughout, including abstract and introduction.

The paper solely focuses on (effective) quadratic problems, so interactions and thus quantum-many-body correlations, while in principle generalizable, are not at all included in the applications. When the summary claims applicability up to millions of degrees of freedom, then the present work is extremely modest only up to a Hilbert space dimension of up to 31,056 states. A few comments in this regard to put things in perspective should be present in the paper.

Along similar lines, when discussing non-interacting models, the terminology of "full bandwidth W" should be clearly defined (it appears, the paper means many-body bandwidth as required by KPM, and not 1-particle bandwidth).

The first application on supercurrent and Josephson inductance is on a system with (50+50+50)*15 = 2250 sites which for a quadratic BdG Hamiltonian can be exactly diagonalized. So a priori, there does not seem to be a need for a hybrid KPM at all. Yet it can serve as a benchmark, which should be emphasized more strongly. Specifically: if all moments were computed (which appears feasible in the first application), then results for current etc. should be exact. It needs to be explained why "KPM only" shows zero contribution to the current operator, given that M=500 is not so far from 2250 sites. So a comparison for different M should be included and discussed (M=500, 1000, 2000, etc.)

Example 2 (double dot Hamiltonian) lacks numerical specifics on the system size simulated. Does the simulation included the view shown in Fig 4? What are the (estimated) finite size effects due to a finite system size in Fig. 4 on the energy resolution in Fig. 5? Fig. 5 only shows the the lowest 4 states which have well defined energies where state 4 already appears with a rather flat dispersion. What about closeby higher lying states 5,6,7,8, .. ? For reference, it would also be of interest to see where the continuum starts.

Example 3: What is the motivation here? (also bearing in mind that the results are not discussed at all). furthermore, it is not quite clear what was done in detail. how was the minimization performed to obtain the effective downfolded model parameters? e.g., what was the cost function? Where does "orbital magnetic field" come into the picture?

Requested changes

see report

---

## Round 2 · Referee Report · Anonymous (Referee 2) · 2019-12-30

Strengths

1- Innovative idea of combining two well-known numerical methods 2- Several model systems have been investigated using the proposed method 3- source code is available online

Weaknesses

1- No interacting systems have been considered 2- Some details needed for readability are missing 3- Comparision to other methods should be extended

Report

The authors propose a method combining sparse diagonalization with the kernel polynomial method (KPM) to numerically study quantum many-body physics. Since both parts of this combination can be performed, by only using matrix-vector multiplications instead of full diagonalization, the method could be interesting for simulating general quantum many-body systems. The idea of combining these two techniques is interesting and the authors choose several applications to demonstrate the power of their approach. The source code is accessible online, which should be very positively noted.

The authors apply the proposed method to three applications. The first application considers a Josephson junction to compute its supercurrent and inductance. The authors show that the hybrid method reproduces the exact result obtained from full diagonalization. It is, however, not clear why the "KPM only" approach fails to reproduce the exact result and vanishes instead. The problem dimension is chosen small, such that full exact diagonalization is feasible. Typically, one would expect sparse methods to be able to reproduce exact results. An explanation for this behavior is not given in the manuscript and needs to be added. In Fig. 1 the current operator spectrum is shown. However, this quantity has not been defined previously and it would improve readability to clearly define this quantity. Also, it is not clear whether the results shown in Figs. 1,2,3 are at zero or finite temperature.

While the chosen problems are interesting research applications, they can be modeled by quadratic Hamiltonians, which do not pose the hardest challenges for numerical methods. It would be interesting to apply their method to interacting problems. The method seems suited to study the low-energy physics of the Hubbard model. While an application of the proposed method is probably beyond the scope of the present manuscript, the authors should comment on whether their method is applicable to study low-energy effective models of interacting systems. A clear discussion of this point would largely benefit the manuscript.

The "stochastic trace approximation" is mentioned in the manuscript. It is not clear whether this method has been used in the three applications. If not, this should be explicitly stated that all traces have been evaluated exactly. Otherwise, I would expect the results to have errorbars.

The authors repeatedly use the term "sparse diagonalization" to compute the extremal eigenvalues. The authors should specify, what algorithm they use for this purpose. Several sparse algorithms, including the widely used Lanczos algorithm, fail at computing many eigenstates, so it would be interesting to have the author's opinion on how to compute multiple extremal eigenstates.

The finite-temperature Lanczos method, as for example proposed in (https://journals.aps.org/prb/abstract/10.1103/PhysRevB.49.5065), is another method capable of performing finite-temperature simulations of quantum many-body systems only using sparse matrix operations. The authors should comment on how their method compares to this rather well-known technique.

The idea proposed by the manuscript eventually deserves publication. However, the authors need to strengthen their case. The authors need to demonstrate clearly how the hybrid method improves upon other algorithms. First, the failure of the "KPM only" method in application A needs to be properly explained. Second, in applications B and C the authors only show that their method is able to reproduce exact results. It would benefit the manuscript if also here, the method was compared to another sparse method.

---

## Editorial Decision

awaiting_resubmission